# In Vitro and In Vivo Human Metabolism of Ostarine, a Selective Androgen Receptor Modulator and Doping Agent

**DOI:** 10.3390/ijms25147807

**Published:** 2024-07-17

**Authors:** Omayema Taoussi, Giulia Bambagiotti, Prince Sellase Gameli, Gloria Daziani, Francesco Tavoletta, Anastasio Tini, Giuseppe Basile, Alfredo Fabrizio Lo Faro, Jeremy Carlier

**Affiliations:** 1Section of Legal Medicine, Department of Biomedical Sciences and Public Health, Marche Polytechnic University, Via Tronto 10/a, 60126 Ancona, Italy; o.taoussi@pm.univpm.it (O.T.); g.bambagiotti@pm.univpm.it (G.B.); p.s.gameli@pm.univpm.it (P.S.G.); g.daziani@pm.univpm.it (G.D.); francescotavoletta@hotmail.it (F.T.); a.tini@pm.univpm.it (A.T.); j.carlier@univpm.it (J.C.); 2Department of Trauma Surgery, IRCCS Galeazzi Orthopedic Institute, Via Riccardo Galeazzi 4, 20161 Milan, Italy; basiletraumaforense@gmail.com

**Keywords:** ostarine, enobasarm, selective androgen receptor modulator, doping, human metabolism, liquid chromatography–high-resolution mass spectrometry (LC-HRMS)

## Abstract

Ostarine (enobasarm) is a selective androgen receptor modulator with great therapeutic potential. However, it is also used by athletes to promote muscle growth and enhance performances without the typical adverse effects of anabolic steroids. Ostarine popularity increased in recent years, and it is currently the most abused “other anabolic agent” (subclass S1.2. of the “anabolic agents” class S1) from the World Anti-Doping Agency’s (WADA) prohibited list. Several cases of liver toxicity were recently reported in regular users. Detecting ostarine or markers of intake in biological matrices is essential to document ostarine use in doping. Therefore, we sought to investigate ostarine metabolism to identify optimal markers of consumption. The substance was incubated with human hepatocytes, and urine samples from six ostarine-positive cases were screened. Analyses were performed via liquid chromatography–high-resolution tandem mass spectrometry (LC-HRMS/MS) and software-assisted data mining, with in silico metabolite predictions. Ten metabolites were identified with hydroxylation, ether cleavage, dealkylation, *O*-glucuronidation, and/or sulfation. The production of cyanophenol-sulfate might participate in the mechanism of ostarine liver toxicity. We suggest ostarine-glucuronide (C_25_H_22_O_9_N_3_F_3_, diagnostic fragments at *m*/*z* 118, 185, and 269) and hydroxybenzonitrile-ostarine-glucuronide (C_25_H_22_O_10_N_3_F_3_, diagnostic fragments at *m*/*z* 134, 185, and 269) in non-hydrolyzed urine and ostarine and hydroxybenzonitrile-ostarine (C_19_H_14_O_4_N_3_F_3_, diagnostic fragments at *m*/*z* 134, 185, and 269) in hydrolyzed urine as markers to document ostarine intake in doping.

## 1. Introduction

Testosterone and related anabolic steroids have been the most abused performance-enhancing drugs for decades. Steroid use is common among amateurs and elite athletes because these substances increase lean body mass, strength, aggressiveness, and speed up recovery after workouts, thereby providing a competitive advantage [1,2,3]. Nevertheless, the adverse effects associated with their use, such as prostate hyperplasia or carcinoma in men and androgenic effects in women, can be troublesome [2,4,5,6,7]. In the recent decade, a new family of drugs called selective androgen receptor modulators (SARMs) has been developed. These substances interfere with the negative feedback loop of the hypothalamic–pituitary–gonadal axis, thus increasing testosterone levels with fewer negative consequences compared to anabolic steroids. SARMs have become prevalent in the performance-enhancing market [8,9]. Consequently, the World Anti-Doping Agency (WADA) has banned SARM use at all times (in and out of competition) since 2008; they are classified as “other anabolic agents” under Section S1.2 of the WADA’s prohibited list [10].

Ostarine, also known as enobasarm, GTx-024, or S-22 [(2S)-3-(4″-cyanophenoxy)-*N*-[4′-cyano-3′-(trifluoromethyl)phenyl]-2-hydroxy-2-methylpropanamide], is a SARM with significant therapeutic potential. It may be used to treat muscle loss conditions such as muscular dystrophy, cancer-associated cachexia, and general sarcopenia, although it is not yet fully approved for clinical use. However, it is also used for doping purposes and is rapidly gaining popularity; it can be easily acquired from various fitness centers or e-vendors as 10 or 25 mg oral tablets, and it is promoted by social media platforms [11,12]. Ostarine abuse, especially among young athletes and bodybuilders, has garnered much attention. Additionally, its unintentional use due to possible contamination or intentional addition to supplements has also been noted [13]. With this growing popularity, cases of ostarine-related liver toxicity have been recently reported in athletes, showing acute cholestatic injury with jaundice and elevated liver enzymes [14,15,16]. Kintz et al. also observed liver cytolysis and massive rhabdomyolysis in a case of long-term exposure to ostarine and cardarine, which is a peroxisome proliferator-activated receptor delta agonist (PPAR-δ) [17].

Identifying ostarine or its consumption markers in an athlete’s sample by a WADA-accredited laboratory can trigger an adverse analytical finding (AAF), potentially leading to sanctions. Ostarine was identified in 306 AAFs reported to WADA’s anti-doping administration and management system from 2016 to 2022 with a steady increase from 0.7% to 1.8% of total AAFs (from 7.0% to 32.4% of AAFs involving S1.2 substances); it is currently the most reported S1.2 substance [18]. The identification of metabolites in biological samples is essential to document exposure in analytical toxicology and doping [19,20,21].

Ostarine metabolism was studied in horse doping [22,23,24], but its pharmacokinetics remain partially understood in humans. In 2010 [25], Thevis et al. assessed ostarine in vitro metabolism using human liver microsomes and S9 fractions; samples were analyzed by liquid chromatography–tandem mass spectrometry (LC-MS/MS), and the elemental composition of the metabolites was verified by LC–high-resolution MS (LC-HRMS). However, there is no clear information on the data mining process, suggesting that unexpected metabolites might have been missed [25]. In 2011, the same research team assessed ostarine metabolism in two urine samples from doping cases, applying the same analytical strategy [26]. They identified seven metabolites previously detected in human liver microsomes and seven additional metabolites produced by hydroxylation, dephenylation, glucuronidation, and/or sulfation. Human liver microsomes do not contain all hepatic enzymes and endogenous cofactors involved in drug metabolism and do not provide the natural orientation of membrane enzymes, with results requiring in vivo confirmation. The authors provided preliminary data on ostarine human metabolism, but the metabolism likely varies based on the individual, the dose administered, the route of administration, and the collection time following exposure [26]. Coss et al. found that ostarine was at little risk of relevant drug–drug interactions with an open-label phase I clinical study [27]. Further studies in humans are necessary to complete these results.

Our objective was to further assess the human metabolism of ostarine using in silico metabolite prediction, in vitro human hepatocyte incubations, and in vivo analysis of ostarine-positive urine samples to identify optimal metabolite markers of consumption. Incubates and samples were analyzed by LC-HRMS/MS and software-assisted data mining for an exhaustive screening of the incubates and samples.

## 2. Results

### 2.1. In Silico Prediction

With GLORYx, 12 (pA1–pA12, by decreasing score) first-generation and 34 second-generation (pAX-1–pAX-6, by decreasing score, pAX as the corresponding first-generation metabolite) metabolites were predicted (Appendix A). Phase II conjugations, especially *O*-glucuronidation at the hydroxyl group and glutathionylation at both nitrile groups, were prominent. Phase I reactions such as amide hydrolysis, *O*-dealkylation, and hydroxylation/oxidation were also predicted. Putative metabolites and associated transformations were used for LC-HRMS/MS analyses (Section 4.6.2) and data mining (Section 4.7), respectively.

### 2.2. Ostarine Fragmentation Pattern

Ostarine was detected only in negative ionization mode in the present analytical conditions. It was eluted at 13.59 min of the chromatographic run with a signal at *m*/*z* 388.0916 (deprotonated ions [M-H]^−^). In HRMS/MS, ether cleavage split the ion in two fragments at *m*/*z* 118.0311 and 269.0549, and amide cleavage produced an additional fragment at *m*/*z* 185.0340, corresponding to the amino-trifluoromethyl-benzonitrile moiety. Ostarine fragmentation pattern in negative ionization mode is displayed in Figure 1.

### 2.3. Metabolite Identification in Human Hepatocyte Incubations

Following incubation with human hepatocytes, nine metabolites were identified and were only detected in negative ionization mode (M1 to M9 by ascending retention time). The extracted-ion chromatogram of ostarine and metabolites after 3 h incubation is displayed in Figure 1. *O*-Glucuronidation and ether cleavage were the main metabolic transformations identified. Other reactions were hydroxylation at the benzonitrile moiety or in position 3 of the propyl chain, dealkylation subsequent to ether cleavage, and sulfation. Results are summarized in Table 1. The fragmentation patterns of ostarine metabolites in negative ionization modes are displayed in Figure 2.

#### 2.3.1. O-Glucuronidation

M6 was the most intense in vitro metabolite and was produced through glucuronidation, as demonstrated by a typical +176.0333 Da shift from the parent (+C_6_H_8_O_6_). The metabolite was eluted at 11.79 min of the LC gradient and presented a base peak at *m*/*z* 564.1249 in negative ionization mode. M6 fragmentation pattern contained the same fragments as ostarine, i.e., *m*/*z* 118.0310, 185.0339, and 269.0544, confirming direct glucuronidation; moreover, it also contained fragments at *m*/*z* 388.0927 due to glucuronide loss, *m*/*z* 445.0916 due to ether cleavage, and *m*/*z* 287.0652 due to ether cleavage on the other side of the oxygen atom and glucuronide loss. Susceptibility to β-glucuronidase hydrolysis (Section 2.4) further indicated that glucuronidation occurred at the hydroxyl group of the molecule.

#### 2.3.2. Ether Cleavage

M2 was eluted at 9.91 min and was produced by benzonitrile loss through ether cleavage (-C_7_H_3_N), as suggested by the −101.0270 Da shift from ostarine. M2 fragment at *m*/*z* 185.0338 indicated that the amino-trifluoromethyl-benzonitrile moiety was not changed; further methylamine loss generated a minor fragment at *m*/*z* 158.0227. Another minor ion at *m*/*z* 257.0534 was produced by methanol loss.

Further dealkylation subsequent to ether cleavage produced metabolite M3, which was eluted at 10.09 min with a base peak at *m*/*z* 257.0544, i.e., a −30.0105 Da shift from M2 (-CH_2_O) and a −131.0370 Da shift from parent (-C_8_H_5_ON). M3 was fragmented to *m*/*z* 185.0338 and 158.0227, indicating that the amino-trifluoromethyl-benzonitrile moiety was not changed.

Ether cleavage at the other side of the oxygen atom generated hydroxybenzonitrile (cyanophenol) (-C_12_H_9_O_2_N_2_F_3_), which was not detected. However, further *O*-sulfation (+SO_3_) produced M1, which was eluted at 4.96 min of the chromatographic gradient with a signal at *m*/*z* 257.0544. In M1 HRMS/MS spectrum, ion at *m*/*z* 118.0310 was produced by sulfate loss.

#### 2.3.3. Sulfation

M7 was produced through ostarine sulfation (+SO_3_), as demonstrated by a +79.9568 Da mass shift from parent, and was eluted at 12.10 min of the LC run. M7 fragments at *m*/*z* 118.0310 and 185.0334, also detected in parent, and fragment at *m*/*z* 388.0927 produced by sulfate loss confirmed the transformation.

#### 2.3.4. Hydroxylation

M8 and M9 were generated by hydroxylation (+O), as shown by the +15.9948/15.9950 Da shift from ostarine; M8 and M9 were eluted at 12.15 and 12.38 min of the gradient, respectively. In M8 HRMS/MS spectrum, fragments at *m*/*z* 118.0310 (hydroxybenzonitrile) and 185.0339 (amino-trifluoromethyl-benzonitrile) indicated that the hydroxylation occurred at the hydroxymethylpropanamide group. More precisely, the fragment at *m*/*z* 255.0393 produced through carbonyl β-cleavage suggested that the transformation occurred in position 3 of the propanamide chain. As opposed to M8, M9 yielded a major fragment at *m*/*z* 134.0258 instead of 118.0310, indicating that the hydroxylation occurred at the benzonitrile moiety of the molecule; fragments at *m*/*z* 185.0339 and 269.0547 confirmed that the other parts were intact.

Further M9 glucuronidation (+C_6_H_8_O_6_) produced M4, which was eluted at 10.33 min with a +192.0276 Da mass shift from parent. M4 fragmentation pattern was similar to that of M4.

Another minor metabolite was M5, which was produced by hydroxylation (+O) and sulfation (+SO_3_), as suggested by the + 95.9516 Da mass shift from parent. Due to low intensity, M5 fragmentation pattern was poor and yielded only two fragments at *m*/*z* 134.0257 through ether cleavage and *m*/*z* 404.0891 through sulfate loss, indicating that the hydroxylation occurred at the benzonitrile group.

### 2.4. Metabolite Identification in Positive Urines

Results are summarized in Table 1. Of the 9 metabolites identified in vitro, 8 were detected in vivo: M7 (ostarine-sulfate) was not detected in urine, whether it was hydrolyzed or not. M6 (ostarine-glucuronide) was the main metabolite in all samples without glucuronide hydrolysis, with the exception of Cases #1 and #3, in which M1 (hydroxybenzonitrile-sulfate) was dominating. However, M1 was also detected in blank urine analyzed in the same conditions, indicating that M1 is not specific and is produced endogenously. Logically, ostarine was the main marker in hydrolyzed urine due to the cleavage of M6 glucuronide, although it was minor in non-hydrolyzed samples. M4 (hydroxybenzonitrile-ostarine-glucuronide) and the corresponding non-conjugated metabolite (M9) were preponderant in non-hydrolyzed and hydrolyzed urine (Cases #2 to #6), respectively. M3 (ether cleavage and dealkylation) was detected in all samples, but only after hydrolysis. The corresponding glucuronide was indeed detected in these samples without hydrolysis, but with an intensity always below the established threshold for reporting. Moreover, an additional minor metabolite, M1a, produced by benzonitrile loss through ether cleavage (-C_7_H_3_N) and further *O*-glucuronidation, was detected in non-hydrolyzed urine from Cases #5 and #6, as suggested by the +75.0053 Da mass shift from the parent and the fragmentation that is similar to that of the corresponding non-conjugated metabolite (M2) (Figure 2). Chromatograms for ostarine and metabolites identified in all samples without hydrolysis are displayed in Figure 3.

## 3. Discussion

The suggested metabolic fate of ostarine is displayed in Figure 4.

The present results were consistent with the studies from Thevis et al., who detected ostarine metabolites with the same transformations in in vitro [25] and in non-hydrolyzed urine samples from two users, 62 h after the oral administration of 25 mg ostarine [26]. The results were also consistent with the study from Walpurgis et al., who followed the signal of specific urinary metabolites after ostarine microdosing in five volunteers [28].

A notable similar result between the study from Thevis et al. in humans [26], the study by Walpurgis et al. [28], and the present results is the predominance of ostarine-glucuronide, although no clear indication was provided on the relative abundance of the metabolites in the article by Thevis et al. [26].

However, differences were also observed. Notably, hydroxybenzonitrile-sulfate (M1) was produced in vitro in the present study. However, the same compound was found in blank urine; therefore, it is not clear whether it was only endogenously produced in vivo or also produced through ostarine metabolism, as observed in hepatocyte incubations. Benzonitrile derivatives are used as herbicides and are highly cytotoxic, as shown in vitro on hepatic cell lines [29]. If hydroxybenzonitrile and its sulfo-conjugated metabolites indeed are produced in vivo through hepatic metabolism, they might participate in ostarine mechanism of liver toxicity in chronic users [14,15,16,17]. More data from regular ostarine users are necessary to test this hypothesis.

Another difference between the study by Thevis et al. [26] and our results is the identification of metabolites hydroxylated at the methyl group of the propanamide chain in the study by Thevis et al.; contrarily, we believe that the fragment at *m*/*z* 255.0387–255.0394 is rather indicative of a hydroxylation in position 3 of the propanamide chain, as reported in our study (Figure 2).

Finally, Thevis et al. [26] also identified 14 metabolites in total, with further metabolization such as dihydroxylation and carboxylation, compared to the present study. This discrepancy might be explained by a different dose and the longer time of collection after ingestion in the study by Thevis et al. A further investigation is necessary to refine these results, especially in long-term users.

Ostarine’s main metabolic transformations in human hepatocytes were *O*-glucuronidation and benzonitrile hydroxylation. We propose ostarine-glucuronide (M6) and hydroxybenzonitrile-ostarine-glucuronide (M4) as specific urinary markers of ostarine use in doping. Alternatively, we suggest the corresponding non-conjugated metabolites, i.e., ostarine itself and hydroxybenzonitrile-ostarine (M9), as the main markers of ostarine consumption in hydrolyzed urine. To the best of our knowledge, these metabolites are not produced by other known endogenous or exogenous molecules. In LC-MS, screening should be performed in negative ionization mode. The detection of these metabolites in biological samples might be crucial to unequivocally prove ostarine use in doping and rule out sample tampering, potentially leading sanctions for the compromised athletes to finally move towards healthier sport competitions.

## 4. Materials and Methods

### 4.1. Chemicals and Reagents

LC-MS-grade acetonitrile (CAS #75-05-08), water (CAS #7732-18-5), methanol (CAS #67-56-1), and formic acid (CAS #64-18-6) were obtained from Carlo Erba (Cornaredo, Italy). Ostarine (CAS #841205-47-8) and diclofenac (CAS #15307-86-5) pure standards were obtained from Toronto Research Chemicals (Toronto, ON, Canada) and Sigma Aldrich (Milan, Italy), respectively. Ostarine and diclofenac stock solutions at 1 mg/mL in methanol were stored at –20 °C before analysis.

Thawing medium and ten-donor-pooled cryopreserved human hepatocytes were obtained from Lonza (Basel, Switzerland). Williams’ medium E, HEPES buffer (2-[4-(2-hydroxyethyl)-1-piperazinyl]ethanesulfonic acid; CAS #7365-45-9), and *l*-glutamine (CAS #56-85-9) were obtained from Sigma Aldrich. Supplemented Williams’ Medium E (SWM) consisted of 2 mmol/L HEPES and 20 mmol/L *l*-glutamine in Williams’ medium E; the solution was stored at 4 °C prior to incubation.

Ostarine-positive urine samples were obtained during drug testing for routine toxicology analyses. The samples tested negative for other doping agents, pharmaceutics, or psychotropic drugs with in-house toxicology screening. This study was conducted in accordance with the Declaration of Helsinki, and informed consent was obtained from the participants.

### 4.2. In Silico Metabolite Prediction

Ostarine human metabolites were predicted with GLORYx software (https://nerdd.univie.ac.at/gloryx/, accesed on 4 July 2024) (University of Hamburg, Germany) [30,31,32]. Ostarine SMILES (simplified molecular input line entry system) generated through ChemSketch (Advanced Chemistry Development, Inc.; v. 2020.1.2) was used to generate a list of putative metabolites; the “phase I and phase II metabolism” option was selected. Metabolites with a score of ≥25% were selected and reprocessed to simulate a second-step transformation; the second-generation metabolite score was multiplied by the corresponding first-generation metabolite score. Metabolites with a prediction score of ≥25% were used for the LC-HRMS/MS inclusion lists and the list of predicted transformations for data mining.

### 4.3. Hepatocyte Incubation

Ostarine incubation with human hepatocytes was performed as previously detailed [32,33,34,35]. Briefly, the hepatocytes were thawed in thawing medium at 37 °C. The medium was then replaced by SWM at 37 °C to reach a concentration of 2 × 10^6^ viable cells/mL. In culture plates, 250 μL of cell suspension in SWM was mixed with 250 μL of 20 μmol/L ostarine in SWM and incubated at 37 °C for 3 h. Negative and positive controls were incubated under the same conditions. The reactions were quenched with 500 μL ice-cold acetonitrile and centrifuged for 10 min at 15,000× *g*. The samples were stored at –80 °C until analysis.

### 4.4. Sample Preparation: Hepatocytes

Incubates were thawed and centrifuged for 10 min, 15,000× *g*, at room temperature. A volume of 100 μL supernatant was mixed with an equal volume of acetonitrile and then centrifuged again in the same conditions. The supernatants were evaporated under a nitrogen stream at 37 °C and reconstituted in 100 μL of mobiles phase A:B 95:5 (*v*/*v*) (Section 4.6.1). After centrifugation in the same conditions, the supernatants were transferred to autosampler vials with a glass insert. A volume of 10 μL was injected into the LC-HRMS/MS.

### 4.5. Sample Preparation: Urine

Samples were thawed at room temperature, and a volume of 100 µL was mixed with 200 µL acetonitrile and centrifuged for 10 min, 15,000× *g*, at room temperature. The supernatants were evaporated to dryness under nitrogen at 37 °C. The supernatants were evaporated under a nitrogen stream at 37 °C and reconstituted in 100 μL of mobiles phase A:B 90:10 (*v*/*v*). After centrifugation in the same conditions, the supernatants were transferred to autosampler vials with a glass insert. A volume of 10 μL was injected into the LC-HRMS/MS.

To further study phase II conjugations, a volume of 100 µL urine was mixed with 10 µL 10 mol/L ammonium acetate, pH 5.0, and 100 µL β-glucuronidase (5000 units), and incubated at 37 °C for 90 min. A volume of 400 µL ice-cold acetonitrile was added, and the samples were mixed and centrifuged for 10 min, 15,000× *g*, at room temperature. The supernatants were evaporated under a nitrogen stream at 37 °C and reconstituted in 100 μL of mobiles phase A:B 90:10 (*v*/*v*). After centrifugation in the same conditions, the supernatants were transferred to autosampler vials with a glass insert. A volume of 10 μL was injected into the LC-HRMS/MS. The samples were simultaneously prepared in the same conditions with 100 µL water instead of enzymes as a negative control.

### 4.6. Instrumental Conditions

LC-HRMS/MS instrumentation was a DIONEX UltiMate 3000 chromatograph coupled with a Q-Exactive mass spectrometer with a heated electrospray ionization source from Thermo Scientific (Waltham, MA, USA).

#### 4.6.1. Liquid Chromatography Conditions

A Kinetex^®^ Biphenyl column (length, 15 cm; diameter, 2.1 mm; particle size, 2.6 μm) from Phenomenex (Torrance, CA, USA) was used for the separation; the column oven temperature was 37 °C. Mobile phases were 0.1% formic acid in water (mobile phase A, MPA) and 0.1% formic acid in acetonitrile (mobile phase B, MPB). The gradient started with 98:2 MPA:B for 2 min, followed by a linear gradient to 35:65 MPA:B within 16 min and a second linear gradient to 5:95 MPA:B within 1 min; 5:95 MPA:B was maintained for 3 min before returning to 98:2 MPA:B within 0.1 min; re-equilibration time was 2.9 min. The MP flow rate was 0.4 mL/min.

#### 4.6.2. Mass Spectrometry Conditions

Ionization conditions were optimized with ostarine standard at 1 μg/mL in MPA:B 90:10 (*v*/*v*). Investigations were performed both in positive and negative ionization mode. HESI source parameters were as follows: sheath gas flow rate at 50 a.u., auxiliary gas flow rate at 5 a.u., spray voltage at ±3.5 kV, capillary and auxiliary gas at 300 °C, and S-lens radio frequency at 50 a.u. Orbitrap calibration was performed for external calibration prior to analysis, and a lock mass list was used for internal calibration for better mass accuracy.

Data were acquired in full-scan HRMS (FullMS)/data-dependent MS/MS (ddMS^2^) mode. FullMS settings were as follows: acquisition range from *m*/*z* 100–750, resolution at 70,000, automatic gain control (AGC) target at 3 × 10^6^, and maximum injection time (IT) at 256 ms. Data dependent settings were as follows: loop count of 5, dynamic exclusion of 2.0 s, and intensity threshold at 10^4^; an inclusion list displayed in Appendix A and based on the in silico predictions was used. ddMS^2^ settings were as follows: isolation window of *m*/*z* 1.2, resolution at 17,500, AGC target at 2 × 10^5^, maximum IT at 64 ms, and normalized collision energy at 20 and 50 a.u.

### 4.7. Metabolite Identification

Raw data were processed with Compound Discoverer (v. 3.1.1.12) from Thermo Scientific, as previously detailed [36]. The list of phase I and phase II transformations based on the in silico predictions is reported in Appendix A. All other parameters were the same as previously reported [34,35].

## 5. Conclusions

The metabolic profile of ostarine, a SARM doping agent, was investigated with ten-donor-pooled human hepatocyte incubations and urine samples from six ostarine-positive cases. A total of ten metabolites produced by *O*-glucuronidation, hydroxylation, ether cleavage, dealkylation, and sulfation were identified with consistent results between in vitro and in vivo data. Ostarine-glucuronide (M6) was the most intense metabolite in all samples. We suggest ostarine-glucuronide and hydroxybenzonitrile-ostarine-glucuronide (M4) in non-hydrolyzed urine and ostarine and hydroxybenzonitrile-ostarine (M9) in hydrolyzed urine as markers to document ostarine intake in doping. The metabolic pattern was consistent with in vivo studies from the scientific literature. A further investigation is necessary to fully understand ostarine metabolism, especially in regular users, and the toxicological relevance of the potential in vivo production of cyanophenol-sulfate.

## Figures and Tables

**Figure 1 ijms-25-07807-f001:**
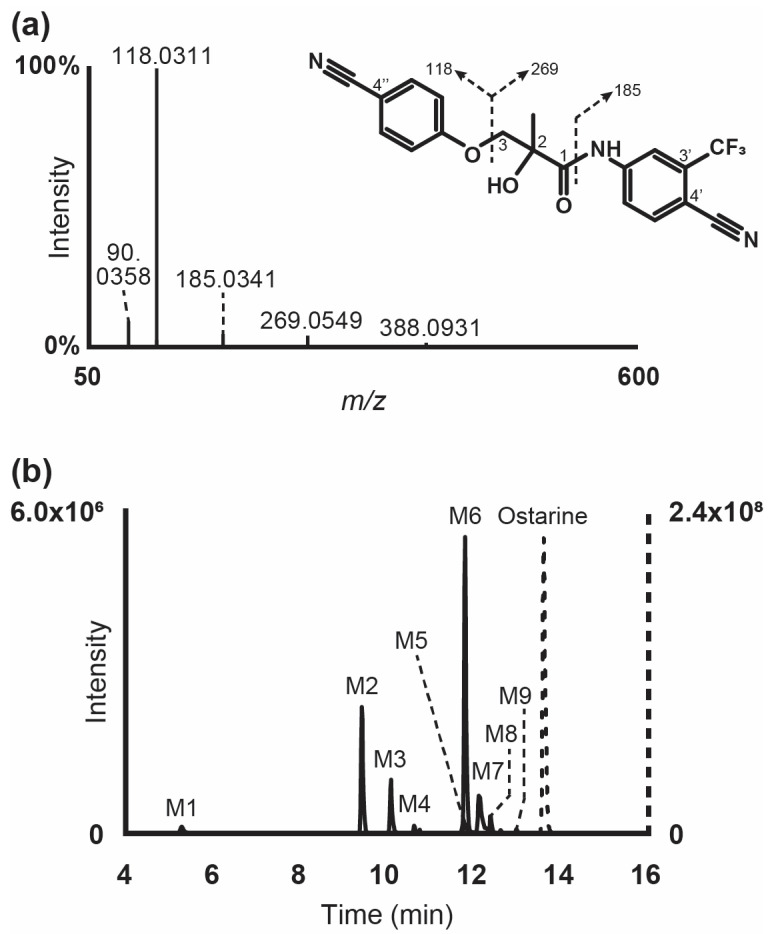
Ostarine high-resolution tandem mass spectrometry spectra after negative electrospray ionization and suggested fragmentation (**a**) and extracted-ion chromatogram in negative ionization mode of ostarine (dashed line, right *y*-axis) and metabolites (plain line, left *y*-axis) after ostarine incubation with 10-donor-pooled human hepatocytes (**b**). Mass tolerance, 5 ppm.

**Figure 2 ijms-25-07807-f002:**
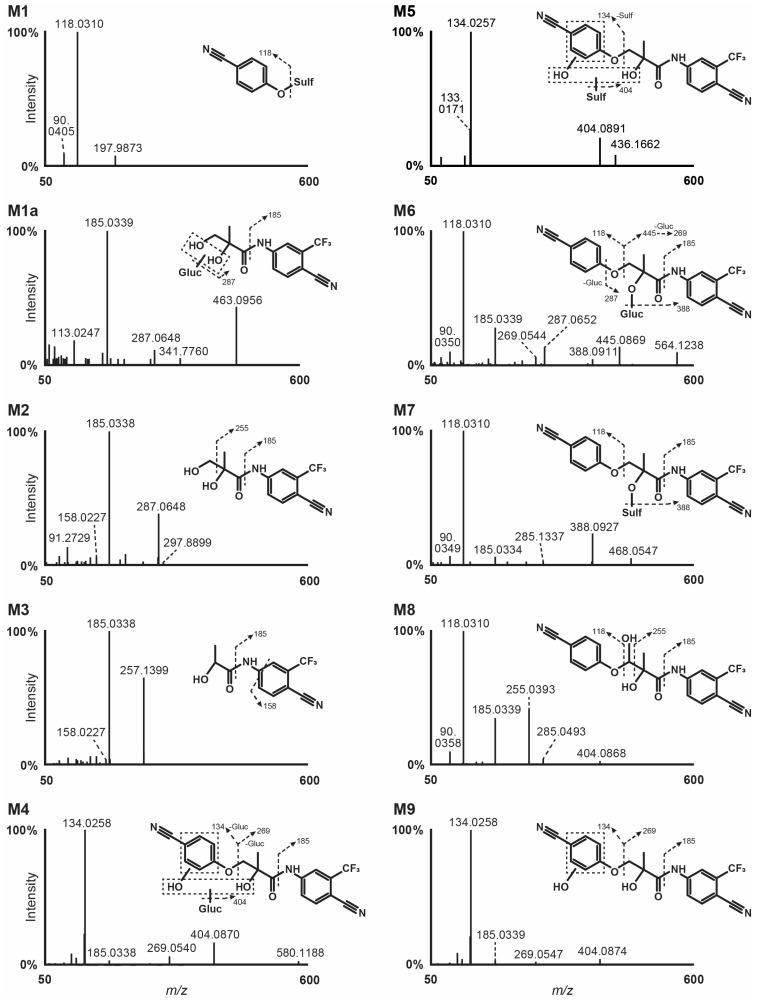
High-resolution tandem mass spectrometry spectra after negative electrospray ionization and suggested fragmentation of ostarine metabolites. Gluc, glucuronide; Sulf, sulfate; dashed box, uncertain position.

**Figure 3 ijms-25-07807-f003:**
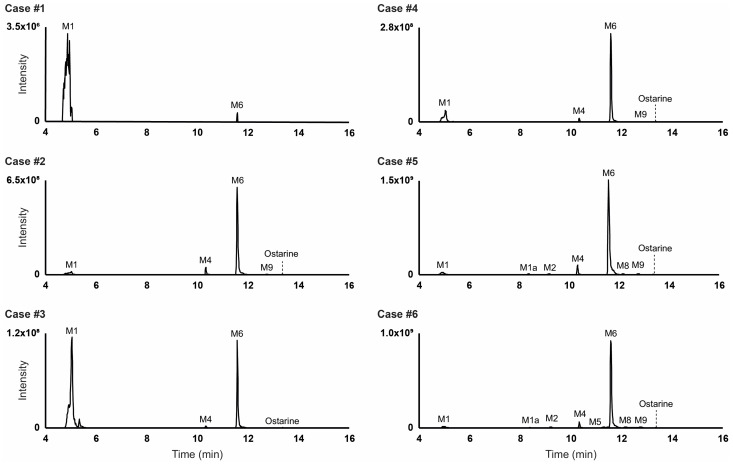
Extracted-ion chromatogram in negative ionization mode of ostarine and metabolites in authentic ostarine-positive urine samples without glucuronide hydrolysis. Mass tolerance, 5 ppm.

**Figure 4 ijms-25-07807-f004:**
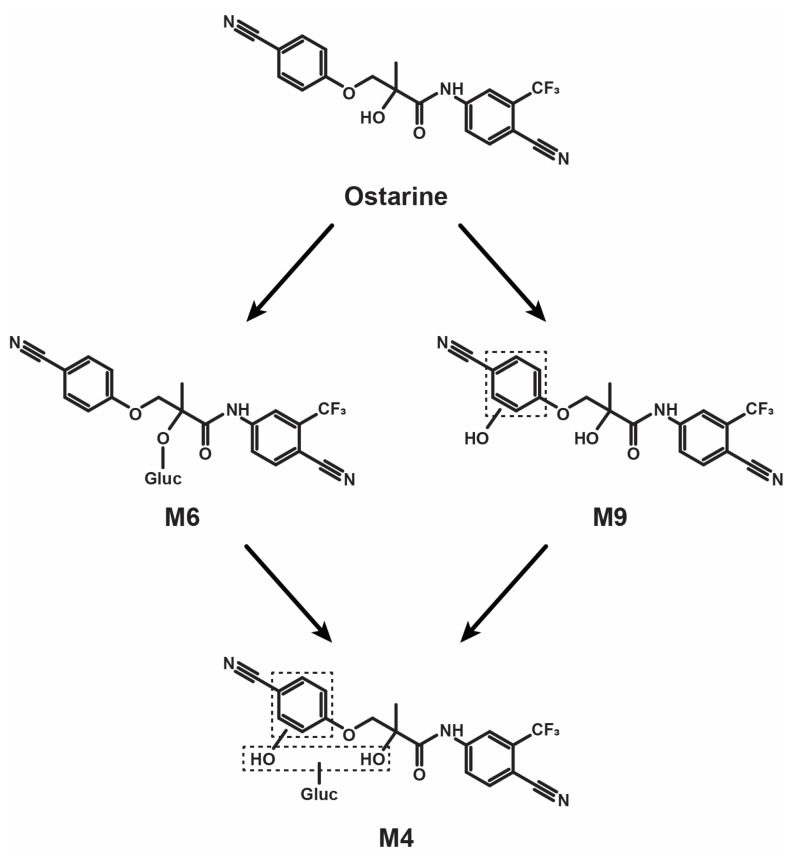
Ostarine suggested metabolic fate in humans (only main metabolites). Gluc, glucuronide; dashed box, uncertain position.

**Table 1 ijms-25-07807-t001:** Ostarine and metabolites identified in human hepatocyte incubations and in ostarine-positive urine samples with (H) or without (wo H) β-glucuronidase hydrolysis. Mass tolerance, 5 ppm; RT, retention time; -, not detected.

ID	Biotransformation	Elemental Composition	RT (min)	*m*/*z* [M-H]^−^ (Δppm)	Hepatocytes	Urine #1 Top: wo H Bottom: H	Urine #2 Top: wo H Bottom: H	Urine #3 Top: wo H Bottom: H	Urine #4 Top: wo H Bottom: H	Urine #5 Top: wo H Bottom: H	Urine #6 Top: wo H Bottom: H
**M1**	Ether Cleavage (2) + Sulfation	C_7_H_5_O_4_NS	4.96	197.9872 (2.77)	7.8 × 10^5^	3.5 × 10^7^ 3.5 × 10^7^	2.3 × 10^8^ 2.2 × 10^8^	8.3 × 10^8^ 1.2 × 10^9^	3.8 × 10^8^ 2.8 × 10^8^	3.8 × 10^8^ 2.4 × 10^8^	1.5 × 10^8^ 1.5 × 10^8^
**M1a**	Ether Cleavage (1) + *O*-Glucuronidation	C_18_H_19_O_9_N_2_F_3_	8.38	463.0969 (−0.20)	-	- -	- -	- -	- -	6.0 × 10^6^ -	6.1 × 10^6^ -
**M2**	Ether Cleavage (1)	C_12_H_11_O_3_N_2_F_3_	9.19	287.0649 (0.00)	8.0 × 10^6^	- -	- 5.2 × 10^6^	- 1.2 × 10^6^	- 6.9 × 10^6^	1.8 × 10^6^ 2.4 × 10^7^	3.2 × 10^6^ 6.0 × 10^7^
**M3**	Ether Cleavage (1) + Dealkylation	C_11_H_9_O_2_N_2_F_3_	9.86	257.0544 (0.25)	3.4 × 10^6^	- -	- 4.1 × 10^7^	- 3.2 × 10^6^	- 1.1 × 10^7^	- 5.2 × 10^7^	- 3.0 × 10^7^
**M4**	Hydroxylation (benzonitrile) + *O*-Glucuronidation	C_25_H_22_O_10_N_3_F_3_	10.33	580.1192 (1.29)	5.0 × 10^5^	- -	1.5 × 10^8^ -	7.4 × 10^6^ -	3.3 × 10^7^ -	4.8 × 10^8^ -	1.8 × 10^8^ -
**M5**	Hydroxylation (benzonitrile) + Sulfation	C_19_H_14_O_7_N_3_F_3_S	11.26	484.0432 (0.04)	1.6 × 10^6^	- -	- -	- -	- -	- -	8.5 × 10^5^ 7.3 × 10^5^
**M6**	*O*-Glucuronidation	C_25_H_22_O_9_N_3_F_3_	11.58	564.1249 (2.40)	1.8 × 10^7^	6.4 × 10^5^ -	2.1 × 10^9^ 7.9 × 10^7^	3.5 × 10^8^ 4.8 × 10^6^	8.5 × 10^8^ 8.6 × 10^7^	7.2 × 10^9^ 3.6 × 10^6^	3.7 × 10^9^ -
**M7**	Sulfation	C_19_H_14_O_6_N_3_F_3_S	12.10	468.0484 (0.29)	4.9 × 10^6^	- -	- -	- -	- -	- -	- -
**M8**	Hydroxylation (3-propyl)	C_19_H_14_O_4_N_3_F_3_	12.15	404.0866 (0.58)	1.1 × 10^6^	- -	- 7.2 × 10^5^	- 4.5 × 10^5^	- 2.4 × 10^6^	2.1 × 10^6^ 1.4 × 10^7^	9.5 × 10^5^ 2.2 × 10^7^
**M9**	Hydroxylation (benzonitrile)	C_19_H_14_O_4_N_3_F_3_	12.73	404.0864 (0.09)	2.6 × 10^5^	- -	1.2 × 10^6^ 4.8 × 10^8^	- 3.9 × 10^7^	2.4 × 10^5^ 1.8 × 10^8^	2.7 × 10^6^ 7.1 × 10^8^	1.9 × 10^6^ 6.1 × 10^8^
Ostarine		C_19_H_14_O_3_N_3_F_3_	13.36	388.0916 (0.40)	1.0 × 10^9^	- 9.5 × 10^6^	6.7 × 10^6^ 4.7 × 10^9^	9.1 × 10^5^ 1.8 × 10^9^	5.7 × 10^6^ 3.5 × 10^9^	3.5 × 10^7^ 8.7 × 10^9^	1.6 × 10^7^ 8.3 × 10^9^

## Data Availability

Raw data were generated at the Department of Biomedical Sciences and Public Health, Marche Polytechnic University. Derived data supporting the findings of this study are available from the corresponding author upon request.

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
