# Peer review of "In Vitro and In Vivo Human Metabolism of Ostarine, a Selective Androgen Receptor Modulator and Doping Agent"

_ijms, 2024, doi:10.3390/ijms25147807_

Round 1
Reviewer 1 Report
Comments and Suggestions for Authors
The article addresses the metabolism of ostarine, a selective androgen receptor modulator (SARM), which is an important topic due to its increasing use as a doping agent. This focus is both timely and relevant for the field of sports medicine and anti-doping research.
The study employs a combination of in vitro and in vivo approaches, including human hepatocyte incubations and the analysis of urine samples from ostarine-positive cases. This dual approach provides a thorough investigation of ostarine metabolism.
The use of liquid chromatography-high-resolution tandem mass spectrometry (LC-HRMS/MS) and in silico metabolite predictions enhances the depth and accuracy of the metabolic analysis.
The article provides detailed identification of ostarine metabolites, including their chemical compositions and the specific metabolic transformations. This level of detail is valuable for both researchers and practitioners in the field.
The study lacks clinical data on the health effects of ostarine use in humans, particularly regarding long-term use and its potential toxicities. Including such data would strengthen the study’s relevance and applicability.
The article uses highly specialized terminology and detailed chemical descriptions that might be difficult for non-experts to follow. Simplifying some of these explanations could broaden the audience.
There are instances of repetition in the introduction and discussion sections, which could be streamlined to improve readability and focus.
While the study identifies specific metabolites as markers for ostarine intake, it provides limited discussion on the practical implications for doping control and public health policies. Expanding this discussion would enhance the article’s impact.
Comments on the Quality of English LanguageQUALITY OF ENGLISH LANGUAGE
The language is generally clear, but some sections are verbose and could be made more concise. The structure of the text is logical, with well-organized sections.
The use of technical terms is appropriate for an academic audience, but some explanations could benefit from simplification to make the content more accessible.
There are minor grammatical errors and punctuation issues, such as inconsistent use of commas and overly long sentences. Addressing these would improve the overall readability.
Consistency in Terminology: The article consistently uses terms related to ostarine metabolism and forensic procedures, ensuring clarity. However, ensuring strict definitions throughout would improve precision.
Line 35: "Testosterone and related anabolic steroids have been the most abused performance-enhancing drugs for decades. Steroid use is not uncommon among amateur athletes and even elites as they increase lean body mass, strength, aggressiveness, and boost recovery after workouts, thereby providing a competitive advantage." Revised: "Testosterone and related anabolic steroids have been the most abused performance-enhancing drugs for decades. Steroid use is common among both amateur and elite athletes because these substances increase lean body mass, strength, aggressiveness, and speed up recovery after workouts, thus providing a competitive advantage."
Line 40: "In the last decade, the discovery and synthesis of a new family of drugs, the selective androgen receptor modulators (SARMs), capable of interfering with the negative feedback loop of the hypothalamic-pituitary-gonadal axis, thereby increasing testosterone levels with lesser negative consequences compared to anabolic steroids, have gained prevalence in the performance enhancing market." Revised: "In the last decade, a new family of drugs called selective androgen receptor modulators (SARMs) has been developed. These drugs interfere with the negative feedback loop of the hypothalamic-pituitary-gonadal axis, increasing testosterone levels with fewer negative consequences compared to anabolic steroids. SARMs have become prevalent in the performance-enhancing market."
Line 49: "Ostarine, also named enobasarm, GTx-024, or S-22 [(2S)-3-(4’’-cyanophenoxy)-N-[4’-cyano-3’-(trifluoromethyl)phenyl]-2-hydroxy-2-methylpropanamide, Fig. 1], is a SARM that has shown tremendous therapeutic potential and may be used for treating conditions manifesting muscle loss such as muscular dystrophy, cancer-associated cachexia, and general sarcopenia, although clinical use was not yet fully approved." Revised: "Ostarine, also known as enobasarm, GTx-024, or S-22, is a SARM with significant therapeutic potential. It may be used to treat muscle loss conditions such as muscular dystrophy, cancer-associated cachexia, and general sarcopenia, although it is not yet fully approved for clinical use."
Line 56: "Ostarine abuse, particularly among young athletes and bodybuilders, and its unintentional use resulting from plausible contamination or intentional addition to supplements has garnered much attention." Revised: "Ostarine abuse, especially among young athletes and bodybuilders, has garnered much attention. Additionally, unintentional use due to possible contamination or intentional addition to supplements has also been noted."
Line 66: "The identification of ostarine or its markers of consumption in an athlete’s sample by a WADA-accredited laboratory can trigger an adverse analytical findings (AAF), potentially leading to sanctions." Revised: "Identifying ostarine or its consumption markers in an athlete’s sample by a WADA-accredited laboratory can trigger an adverse analytical finding (AAF), potentially leading to sanctions."
Line 94: "Our objective was to further assess ostarine human metabolism using in silico metabolite prediction, in vitro human hepatocyte incubations, and in vivo analysis of ostarine-positive urine samples to identify optimal metabolite markers of consumption." Revised: "Our objective was to further assess the human metabolism of ostarine using in silico metabolite prediction, in vitro human hepatocyte incubations, and in vivo analysis of ostarine-positive urine samples to identify optimal markers of consumption."
Line 213: "Ostarine suggested metabolic fate is displayed in Figure 4. The present results were consistent with the study from Thevis et al. [26,28], who detected ostarine metabolites with the same transformations in non-hydrolyzed urine samples of two users, 62 h after the oral administration of 25 mg ostarine." Revised: "The suggested metabolic fate of ostarine is displayed in Figure 4. The present results are consistent with the study by Thevis et al. [26,28], who detected ostarine metabolites with the same transformations in non-hydrolyzed urine samples from two users, 62 hours after the oral administration of 25 mg of ostarine."
Line 214: "The present results were consistent with the study from Thevis et al. [26,28], who detected ostarine metabolites with the same transformations in non-hydrolyzed urine samples of two users, 62 h after the oral administration of 25 mg ostarine." Revised: "The present results are consistent with the study by Thevis et al. [26,28], who detected ostarine metabolites with the same transformations in non-hydrolyzed urine samples from two users, 62 hours after the oral administration of 25 mg of ostarine."
By applying these corrections, the text will become clearer, more concise, and easier to understand, while maintaining the necessary technical detail for a specialized audience. This will improve the overall readability and impact of the article.
Author Response
The article addresses the metabolism of ostarine, a selective androgen receptor modulator (SARM), which is an important topic due to its increasing use as a doping agent. This focus is both timely and relevant for the field of sports medicine and anti-doping research.
The study employs a combination of in vitro and in vivo approaches, including human hepatocyte incubations and the analysis of urine samples from ostarine-positive cases. This dual approach provides a thorough investigation of ostarine metabolism.
The use of liquid chromatography-high-resolution tandem mass spectrometry (LC-HRMS/MS) and in silico metabolite predictions enhances the depth and accuracy of the metabolic analysis.
The article provides detailed identification of ostarine metabolites, including their chemical compositions and the specific metabolic transformations. This level of detail is valuable for both researchers and practitioners in the field.
- The study lacks clinical data on the health effects of ostarine use in humans, particularly regarding long-term use and its potential toxicities. Including such data would strengthen the study’s relevance and applicability.
Response: We agree with the suggestion and modified the text as follows:
L57-61: “With this growing popularity, cases of ostarine-related liver toxicity have been recently reported in athletes, showing acute cholestatic injury with jaundice and elevated liver enzymes [15–17]. Kintz et al. also observed liver cytolysis and massive rhabdomyolysis in a case of long-term exposure to ostarine and cardarine, a peroxisome proliferator-activated receptor delta agonist (PPAR-δ) [18].”
- The article uses highly specialized terminology and detailed chemical descriptions that might be difficult for non-experts to follow. Simplifying some of these explanations could broaden the audience.
Response: We understand that the reviewer refers to the Results section that describes the metabolites’ fragmentation to elucidate their structure, which indeed contains technical/chemical terms that can be difficult to understand for the layman. However, these explanations are actually intended for analytical toxicologists and chemists who want to use the present results to update their detection methods, and we therefore believe that the technical terms should be used for better accuracy. For a much easier understanding, the fragmentations described in the text were also reported in Figures 1 and 2.
- There are instances of repetition in the introduction and discussion sections, which could be streamlined to improve readability and focus.
Response: We agree with the reviewer. The manuscript was revised to avoid such repetitions.
- While the study identifies specific metabolites as markers for ostarine intake, it provides limited discussion on the practical implications for doping control and public health policies. Expanding this discussion would enhance the article’s impact.
Response: As suggested, the discussion was modified as follows:
L242-245: “The detection of these metabolites in biological samples might be crucial to unequivocally prove ostarine use in doping and rule out sample tampering, potentially leading sanctions for the compromised athletes to finally move towards healthier sport competitions.”
- The language is generally clear, but some sections are verbose and could be made more concise. The structure of the text is logical, with well-organized sections.
Response: We agree with the suggestion, and the manuscript was thoroughly reviewed for better reading.
- The use of technical terms is appropriate for an academic audience, but some explanations could benefit from simplification to make the content more accessible.
Response: See reply to comment #2.
- There are minor grammatical errors and punctuation issues, such as inconsistent use of commas and overly long sentences. Addressing these would improve the overall readability. Consistency in Terminology: The article consistently uses terms related to ostarine metabolism and forensic procedures, ensuring clarity. However, ensuring strict definitions throughout would improve precision.
Response: We thoroughly reviewed the manuscript to address these issues.
- Line 35: "Testosterone and related anabolic steroids have been the most abused performance-enhancing drugs for decades. Steroid use is not uncommon among amateur athletes and even elites as they increase lean body mass, strength, aggressiveness, and boost recovery after workouts, thereby providing a competitive advantage." Revised: "Testosterone and related anabolic steroids have been the most abused performance-enhancing drugs for decades. Steroid use is common among both amateur and elite athletes because these substances increase lean body mass, strength, aggressiveness, and speed up recovery after workouts, thus providing a competitive advantage."
Response: The sentence was modified as suggested.
- Line 40: "In the last decade, the discovery and synthesis of a new family of drugs, the selective androgen receptor modulators (SARMs), capable of interfering with the negative feedback loop of the hypothalamic-pituitary-gonadal axis, thereby increasing testosterone levels with lesser negative consequences compared to anabolic steroids, have gained prevalence in the performance enhancing market." Revised: "In the last decade, a new family of drugs called selective androgen receptor modulators (SARMs) has been developed. These drugs interfere with the negative feedback loop of the hypothalamic-pituitary-gonadal axis, increasing testosterone levels with fewer negative consequences compared to anabolic steroids. SARMs have become prevalent in the performance-enhancing market."
Response: The sentence was modified as suggested.
- Line 49: "Ostarine, also named enobasarm, GTx-024, or S-22 [(2S)-3-(4’’-cyanophenoxy)-N-[4’-cyano-3’-(trifluoromethyl)phenyl]-2-hydroxy-2-methylpropanamide, Fig. 1], is a SARM that has shown tremendous therapeutic potential and may be used for treating conditions manifesting muscle loss such as muscular dystrophy, cancer-associated cachexia, and general sarcopenia, although clinical use was not yet fully approved." Revised: "Ostarine, also known as enobasarm, GTx-024, or S-22, is a SARM with significant therapeutic potential. It may be used to treat muscle loss conditions such as muscular dystrophy, cancer-associated cachexia, and general sarcopenia, although it is not yet fully approved for clinical use."
Response: The sentence was modified as suggested.
- Line 56: "Ostarine abuse, particularly among young athletes and bodybuilders, and its unintentional use resulting from plausible contamination or intentional addition to supplements has garnered much attention." Revised: "Ostarine abuse, especially among young athletes and bodybuilders, has garnered much attention. Additionally, unintentional use due to possible contamination or intentional addition to supplements has also been noted."
Response: The sentence was modified as suggested.
- Line 66: "The identification of ostarine or its markers of consumption in an athlete’s sample by a WADA-accredited laboratory can trigger an adverse analytical findings (AAF), potentially leading to sanctions." Revised: "Identifying ostarine or its consumption markers in an athlete’s sample by a WADA-accredited laboratory can trigger an adverse analytical finding (AAF), potentially leading to sanctions."
Response: The sentence was modified as suggested.
- Line 94: "Our objective was to further assess ostarine human metabolism using in silico metabolite prediction, in vitro human hepatocyte incubations, and in vivo analysis of ostarine-positive urine samples to identify optimal metabolite markers of consumption." Revised: "Our objective was to further assess the human metabolism of ostarine using in silico metabolite prediction, in vitro human hepatocyte incubations, and in vivo analysis of ostarine-positive urine samples to identify optimal markers of consumption."
Response: The sentence was modified as suggested.
- Line 213: "Ostarine suggested metabolic fate is displayed in Figure 4. The present results were consistent with the study from Thevis et al. [26,28], who detected ostarine metabolites with the same transformations in non-hydrolyzed urine samples of two users, 62 h after the oral administration of 25 mg ostarine." Revised: "The suggested metabolic fate of ostarine is displayed in Figure 4. The present results are consistent with the study by Thevis et al. [26,28], who detected ostarine metabolites with the same transformations in non-hydrolyzed urine samples from two users, 62 hours after the oral administration of 25 mg of ostarine."
Response: The sentence was modified as suggested.
- Line 214: "The present results were consistent with the study from Thevis et al. [26,28], who detected ostarine metabolites with the same transformations in non-hydrolyzed urine samples of two users, 62 h after the oral administration of 25 mg ostarine." Revised: "The present results are consistent with the study by Thevis et al. [26,28], who detected ostarine metabolites with the same transformations in non-hydrolyzed urine samples from two users, 62 hours after the oral administration of 25 mg of ostarine."
Response: The sentence was modified as suggested.
By applying these corrections, the text will become clearer, more concise, and easier to understand, while maintaining the necessary technical detail for a specialized audience. This will improve the overall readability and impact of the article.
Reviewer 2 Report
Comments and Suggestions for Authors
Line 19, typo “in”
Lines 322 – 323, unit missing for gas flow rates
Figure 3 – is the scale on which the ostarine is the same as the metabolites or is it a similar scenario to Figure 1b where it has a separate scale/axis?
3. Discussion – a few studies by Thevis et al were cited. It would be clearer to add the specific reference number after the citing sentence to indicate the exact article(s) being cited.
6. Reference – there seem to be some mix-ups with the references and their numbers: References 25-28. Eg. Line 214 refers to Thevis et al but the numbers given are for different articles.
Overall, a concise, clear manuscript of good quality. The study design is simple yet effective and the data presented bridges the gap of previous studies of the same (or similar type of) molecule. A suggestion to improve the thoroughness of the manuscript is to include some discussion regarding the possibility (or the lack thereof) of the two suggested metabolite biomarkers, M4 and M6, being from sources other than the consumption of ostarine.
Author Response
- Line 19, typo “in”
Response: The typo was corrected.
- Lines 322 – 323, unit missing for gas flow rates
Response: With this instrument, gas flow rates are indicated in arbitrary units. The sentence was modified for better clarity:
L318-320: “… sheath-gas flow rate at 50 a.u., auxiliary-gas flow rate at 5 a.u., spray voltage at ±3.5 kV, capillary and auxiliary gas at 300°C, and S-lens radio frequency at 50 a.u..”
- Figure 3 – is the scale on which the ostarine is the same as the metabolites or is it a similar scenario to Figure 1b where it has a separate scale/axis?
Response: In hepatocyte incubates (Fig 1b), the intensity of the parent is not relevant because in excess, and this is why we used a different scale for ostarine. In Figure 3, ostarine and metabolites all use the same scale. We modified the legend of Figure 1 for better clarity:
L111-112: “… extracted-ion chromatogram in negative-ionization mode of ostarine (right y-axis) and metabolites (left y-axis) after ostarine incubation…”
- Discussion – a few studies by Thevis et al were cited. It would be clearer to add the specific reference number after the citing sentence to indicate the exact article(s) being cited.
Response: As suggested, the text was modified as follows:
L207-215: “The present results were consistent with the studies from Thevis et al., who detected ostarine metabolites with the same transformations in vitro [26] and in non-hydrolyzed urine samples from two users, 62 h after the oral administration of 25 mg ostarine [27]. The results were also consistent with the study from Walpurgis et al., who followed the signal of specific urinary metabolites after ostarine microdosing in five volunteers [29]. A notable similar result between the study from Thevis et al. in humans [27], the study by Walpurgis et al. [29], and the present results is the predominance of ostarine-glucuronide, although no clear indication was provided on the relative abundance of the metabolites in the article by Thevis et al.”
L225-230: “Another difference between the study by Thevis et al. [27] and our results is the identification of metabolites hydroxylated at the methyl group of the propanamide chain in the study by Thevis et al.; contrarily, we believe that the fragment at m/z 255.0387-255.0394 is rather indicative of a hydroxylation in position 3 of the propanamide chain, as reported in our study (Fig. 2). Finally, Thevis et al. [27] also …”
- Reference – there seem to be some mix-ups with the references and their numbers: References 25-28. Eg. Line 214 refers to Thevis et al but the numbers given are for different articles.
Response: References 27 and 29 (26 and 28 at the previous submission) are from Thevis et al., but we indeed forgot to mention the in vitro study from the same research group (reference 26; re 25 at the previous submission). After carefully checking the references, there were no mix-ups. Reference 26 was added to the sentence.
- Overall, a concise, clear manuscript of good quality. The study design is simple yet effective and the data presented bridges the gap of previous studies of the same (or similar type of) molecule. A suggestion to improve the thoroughness of the manuscript is to include some discussion regarding the possibility (or the lack thereof) of the two suggested metabolite biomarkers, M4 and M6, being from sources other than the consumption of ostarine.
Response: As suggested, we modified the text as follows:
L240-241: “To the best of our knowledge, these metabolites are not produced by other known endogenous or exogenous molecules.”
Reviewer 3 Report
Comments and Suggestions for Authors
Authors proposed a paper entitled “In vitro and in vivo human metabolism of ostarine, a selective androgen receptor modulator (SARM) and doping agent” for the publication in IJMS.
The paper has a good scientific soundness and deserves to be published after minor revisions.
I suggest the introduction of an abbreviation list, including AAF, WADA, MS, etc.
Line 17. “other” why “other”? Does this represent a final classification under a specific class ?
Line 29. “Further investigation is necessary to fully understand ostarine metabolism, especially inregular users.” I would remove this sentence from the abstract and maybe add it to the conclusion section. Instead, it would be better to add some more results in the final part of the abstract, even numbers in order to increase the hype on the paper.
Line 40-45. It would be better to divide this sentence, that appears to be too long to the readers.
Line 54. “It can” please avoid capital letter to “It”.
Figure 1b. Is this chromatogram made by the authors or has been taken from the literature? In the last case, it should be reported the permission paragraph. In the first case, instead, it should be moved at least to Methods section, or even to Results section.
Line 76. “in vitro” should be reported in italics.
Concerning reactant declared in the Materials method, please add CAS numbers where possible (example, acetonitrile).
Further perspective can be added to the last conclusive paragraph, as a final sentence or period.
Comments on the Quality of English LanguageEnglish quality is quite good.
Author Response
Authors proposed a paper entitled “In vitro and in vivo human metabolism of ostarine, a selective androgen receptor modulator (SARM) and doping agent” for the publication in IJMS.
The paper has a good scientific soundness and deserves to be published after minor revisions.
- I suggest the introduction of an abbreviation list, including AAF, WADA, MS, etc.
Response: We agree that the number of abbreviations can be confusing. However, the journal’s template does not contain a specific section for this purpose.
- Line 17. “other” why “other”? Does this represent a final classification under a specific class ?
Response: “Other anabolic agents” is a subclass of the WADA “anabolic agents” class. For better understanding, we modified the text as follows:
L16-17: “… is currently the most abused “other anabolic agent” (subclass S1.2. of the “anabolic agents” class S1) from the World Anti-Doping Agency (WADA) prohibited list.”
- Line 29. “Further investigation is necessary to fully understand ostarine metabolism, especially in regular users.” I would remove this sentence from the abstract and maybe add it to the conclusion section. Instead, it would be better to add some more results in the final part of the abstract, even numbers in order to increase the hype on the paper.
Response: We agree with the suggestion and modified the text as follows:
L26-30: “We suggest ostarine-glucuronide (C25H22O9N3F3, diagnostic fragments at m/z 118, 185, and 269) and hydroxybenzonitrile-ostarine-glucuronide (C25H22O10N3F3, diagnostic fragments at m/z 134, 185, and 269) in non-hydrolyzed urine, and ostarine and hydroxybenzonitrile-ostarine (C19H14O4N3F3, diagnostic fragments at m/z 134, 185, and 269) in hydrolyzed urine, as markers to document ostarine intake in doping.”
- Line 40-45. It would be better to divide this sentence, that appears to be too long to the readers.
Response: The sentence was split for better readability:
L40-45: “In the last decade, a new family of drugs called selective androgen receptor modulators (SARMs) has been developed. These substances interfere with the negative feedback loop of the hypothalamic-pituitary-gonadal axis, thereby increasing testosterone levels with fewer negative consequences compared to anabolic steroids. SARMSs have become prevalent in the performance-enhancing market [9,10].”
- Line 54. “It can” please avoid capital letter to “It”.
Response: The typo was corrected.
- Figure 1b. Is this chromatogram made by the authors or has been taken from the literature? In the last case, it should be reported the permission paragraph. In the first case, instead, it should be moved at least to Methods section, or even to Results section.
Response: The chromatogram and the MS/MS were generated with our results. As suggested, the figure was moved to the results section.
- Line 76. “in vitro” should be reported in italics.
Response: “in vitro” was reported in italics at this line and also at another occurrence in the text (section 2.3.1.).
- Concerning reactant declared in the Materials method, please add CAS numbers where possible (example, acetonitrile).
Response: As suggested, CAS numbers were added where applicable.
- Further perspective can be added to the last conclusive paragraph, as a final sentence or period.
Response: As suggested, we added the following sentence to the conclusion:
L355-358: “Further investigation is necessary to fully understand ostarine metabolism, especially in regular users, and the toxicological relevance of the potential in vivo production of cyanophenol-sulfate.”
Reviewer 4 Report
Comments and Suggestions for Authors
In the manuscript submitted to me for review entitled "In vitro and in vivo human metabolism of ostarine, a selective androgen receptor modulator (SARM) and doping agent“ the authors Omayema Taoussi, Giulia Bambagiotti, Prince Sellase Gameli, Gloria Daziani, Francesco Tavoletta, Anastasio Tini, Giuseppe Basile, Angelo Montana, Alfredo Fabrizio Lo Faro and Jeremy Carlier present a study investigating the metabolism of ostarine (enobasarm) in vivo and in vitro. In addition to its great therapeutic potential, ostarine is currently the most used "other anabolic agent" and is included in the prohibited list of substances by the World Anti-Doping Agency (WADA).
The present study extends the possibility of detecting different metabolic products obtained after the consumption of ostarine and thus may increase the possibility of the detection of ostarine or markers for its uptake in biological material.
The methods presented by the authors are well selected and described and can be reproduced in a subsequent study. The obtained results are presented in detail using 4 figures, 1 table and three supplementary tables presented in the supplementary file. The conclusions drawn by the authors are well-formed and represent an accurate summary of the results presented.
To support their research, the authors used 36 references, which present information from studies published mostly in the last two decades, but some data from older studies are also presented. More than 1/2 of the total references are from the last 5 years, indicating that the use and detection of various substances (mostly in professional athletes) is a relatively new and current topic and would be of interest to IJMS readers. I did not notice any redundant self-citations, all references used are appropriate and necessary for the preparation of the manuscript.
My remarks and recommendations to the authors are:
1. It is indicated that hepatocytes taken from donors were used in the research. It would be good for the scientific value of the manuscript if even briefly the method of their isolation is added.
2. Has consent been obtained from the hepatocyte donors for taking biological material from them? If there is, let it be indicated in the manuscript.
3. Is there permission from any ethical commission or organization to conduct the in vivo experimental part? At least I didn't notice it noted anywhere.
Author Response
In the manuscript submitted to me for review entitled "In vitro and in vivo human metabolism of ostarine, a selective androgen receptor modulator (SARM) and doping agent“ the authors Omayema Taoussi, Giulia Bambagiotti, Prince Sellase Gameli, Gloria Daziani, Francesco Tavoletta, Anastasio Tini, Giuseppe Basile, Angelo Montana, Alfredo Fabrizio Lo Faro and Jeremy Carlier present a study investigating the metabolism of ostarine (enobasarm) in vivo and in vitro. In addition to its great therapeutic potential, ostarine is currently the most used "other anabolic agent" and is included in the prohibited list of substances by the World Anti-Doping Agency (WADA).
The present study extends the possibility of detecting different metabolic products obtained after the consumption of ostarine and thus may increase the possibility of the detection of ostarine or markers for its uptake in biological material.
The methods presented by the authors are well selected and described and can be reproduced in a subsequent study. The obtained results are presented in detail using 4 figures, 1 table and three supplementary tables presented in the supplementary file. The conclusions drawn by the authors are well-formed and represent an accurate summary of the results presented.
To support their research, the authors used 36 references, which present information from studies published mostly in the last two decades, but some data from older studies are also presented. More than 1/2 of the total references are from the last 5 years, indicating that the use and detection of various substances (mostly in professional athletes) is a relatively new and current topic and would be of interest to IJMS readers. I did not notice any redundant self-citations, all references used are appropriate and necessary for the preparation of the manuscript.
My remarks and recommendations to the authors are:
- It is indicated that hepatocytes taken from donors were used in the research. It would be good for the scientific value of the manuscript if even briefly the method of their isolation is added.
Response: The pooled hepatocytes were directly bought from Lonza (Basel, Switzerland), and unfortunately, their isolation protocol is not disclosed.
- Has consent been obtained from the hepatocyte donors for taking biological material from them? If there is, let it be indicated in the manuscript.
Response: As the hepatocytes were directly bought from a private company (Lonza; Basel, Switwerland) and anonymized upon reception, informed consent is not required for this study.
- Is there permission from any ethical commission or organization to conduct the in vivo experimental part? At least I didn't notice it noted anywhere.
Response: As the hepatocytes were directly bought from a private company (Lonza; Basel, Switwerland) and anonymized upon reception, ethics committee approval is not required for this study.